# Predictors of Youth Accessibility for a Mobile Phone-Based Life Skills Training Program for Addiction Prevention

**DOI:** 10.3390/ijerph20146379

**Published:** 2023-07-17

**Authors:** Severin Haug, Nikolaos Boumparis, Andreas Wenger, Michael Patrick Schaub, Nikolai Kiselev

**Affiliations:** Swiss Research Institute for Public Health and Addiction, Zurich University, Konradstrasse 32, 8005 Zurich, Switzerland; niko.boumparis@isgf.uzh.ch (N.B.); andreas.wenger@isgf.uzh.ch (A.W.); michael.schaub@isgf.uzh.ch (M.P.S.); nikolai.kiselev@isgf.uzh.ch (N.K.)

**Keywords:** addiction, prevention, students, adolescents, mobile phone, predictors

## Abstract

Background: Digital interventions are an emerging and promising avenue for addiction prevention and mental health promotion, but their reach and use are often limited, and little is known about the factors associated with youth accessibility. *SmartCoach* is a life skills training program for addiction prevention where adolescents are proactively invited for program participation in secondary school classes. The mobile phone-based program provides individualized coaching for a period of 4 months and addresses self-management skills, social skills, and substance use resistance skills. This study examined sociodemographic and other predictors of program participation and program use. Methods: A total of 476 adolescents in 28 secondary and upper secondary school classes in the German-speaking part of Switzerland were proactively invited for participation in the *SmartCoach* program. Using generalized linear mixed models (GLMMs), we examined predictors of both program participation and program use at the individual and school class levels. Results: In total, 315 (66.2%) of the present 476 adolescents gave their active consent and provided the necessary information to be included in the program. None of the individual sociodemographic characteristics significantly predicted program participation, however, the participation rate was significantly higher in upper secondary school classes (84%) than secondary school classes (59%). The mean number of interactions with the program was 15.9, i.e., participants took part in almost half of the 34 possible interactions with the *SmartCoach* program. None of the baseline characteristics on the level of the school class significantly predicted program use. On the level of the individual, the univariate models showed that, compared to the reference category of 14-year-old students, program use was significantly lower for students who were 16 or older. Furthermore, participants with a migration background or an origin from a non-German-speaking country showed significantly lower program use. Finally, students with a medium level of perceived stress showed higher program use compared to those with a low level of stress. Within the final multivariate model for program use, only the variable “origin from a non-German-speaking country” remained significant. Conclusions: *SmartCoach* is an attractive offer for young people, in which two out of three young people who are invited in the classroom to participate do so. Among the program participants, the use of the program is acceptable, with an average of almost half of the content being worked on. There is potential for improvement in terms of recruitment, especially in school classes with a lower level of education. The most important starting point for improving program use lies in taking greater account of needs and wishes of students with non-German-speaking countries of origin.

## 1. Introduction

Although there is increasing recognition of social inequalities in adolescent health [1] and socially stratifying factors are robust predictors of health disparities, there is little evidence on the equity impact of interventions in this population group [2,3]. The equity impact of an intervention refers to the extent to which the intervention affects different groups of people equitably, regardless of their socioeconomic status, race, ethnicity, gender, or other characteristics. Based on definitions by Brown et al. [4], the equity impact of an intervention can be (1) positive, if lower socioeconomic status (SES) groups are relatively more responsive to the intervention, (2) neutral, if there is no social gradient in the effectiveness of the intervention, (3) negative, if there is evidence that higher SES groups are relatively more responsive to the intervention, or (4) mixed, if the effect of the intervention varies by SES measure. A review of systematic reviews that addressed socioeconomic inequalities and the equity impact of population-level interventions for adolescent health [3] showed that less than a third of the 140 reviews considered reported differential intervention impact, 13% described participants using a measure of socioeconomic status (SES), and 11% reported differential intervention effects.

The Reach Effectiveness Adoption Implementation Maintenance (RE-AIM) framework is a planning and evaluation framework that was developed to help make research findings more generalizable by encouraging scientists and evaluators to balance internal and external validity when developing and testing interventions [5,6]. Based on this framework, major dimensions which determine the public health impact of an intervention and could be affected by intervention-generated social inequalities include the reach of the target population and its effectiveness. While differential effectiveness or moderators of outcomes are more frequently reported, differential reach or accessibility of the target group is often widely unknown, particularly in digital intervention programs where recruitment often takes place online and the recruitment channels are difficult to trace. This lack of knowledge about the reached compared to the eligible group of people also results in a potentially limited representativeness of the reported results. Therefore, strengthening the evidence on intervention-generated inequalities, including accessibility, remains a priority for public health research [3]. Accessibility or reach of the program includes two major dimensions, which were considered in this study: (1) program participation refers to the act of being involved or enrolled in the program and (2) program use refers to the actual utilization or interaction with the program. The analysis of predictors of program participation is particularly suitable and feasible for school-based interventions, where there is a clearly definable target group and individuals can be interviewed within the framework of compulsory education. Schmid et al. [7] investigated individual and school class-specific characteristics that influence the willingness to participate in a text messaging-based program to promote smoking cessation among adolescents at vocational schools in Switzerland. The participation rate in the program among the eligible smokers present in the classroom was 75%. Concerning sociodemographic factors, the study revealed that having a migration background reduced the willingness to participate. Furthermore, daily compared to occasional smoking and a higher intention to quit smoking increased the willingness to participate. At the class level, a larger number of people present and an earlier time of day of the invitation to participate in the program increased the willingness to participate. Thrul et al. [8] investigated which individual characteristics promoted participation in a group smoking cessation program among adolescents recruited from 42 German secondary school classes. While demographic variables did not influence participation in the smoking cessation program, previous smoking attempts, higher nicotine dependence, and higher smoking cessation motivation were positively associated with willingness to participate.

Haug et al. [9] investigated individual and class-specific characteristics that influenced the willingness to participate in an Internet- and text messaging-based program to reduce binge drinking among vocational and upper secondary school students in Switzerland. The participation rate in the program among the present students in the classroom was 74%. At the class level, a smaller number of present students and a lower proportion of persons with a migration background were associated with a higher willingness to participate in the program; at the individual level, these factors were a female gender, a lower age, and a higher maximum alcohol consumption in the last month.

A web- and text-messaging-based life skills program for vocational school students in Switzerland, with a participation rate among the present students in the classroom of 81%, showed that female gender and occasional binge drinking were positively associated with willingness to participate, while an immigrant background and tobacco smoking were negatively associated [10].

A recent Australian study investigated the uptake of an app-based health promotion program among adolescents in 71 secondary schools. Among 2489 students who had the ability and opportunity to access the Health4Life app, 407 (16.4%) accessed it [11]. Factors associated with program uptake were teacher prompts, living in a major city, and being female. Psychological distress was not a significant predictor of likelihood to access the app, nor was SES or number of health risk behaviors.

Beyond program participation, use or engagement is a major dimension of program reach which could determine the effectiveness and public health impact of an intervention [12]. Reviews of digital interventions for mental health promotion [13,14] or substance use prevention [15] among young people point to the relatively low levels of user engagement. However, only a few studies have examined the predictors of engagement in digital interventions for mental health or substance use among young people. In an online depression prevention program for young adolescents in Australian public schools, lower program engagement was predicted by being older, living in an urban area, lower levels of depressive symptoms, and lower self-esteem at baseline [16]. In a text messaging-based smoking cessation intervention aimed at adolescents from vocational and secondary schools in Switzerland, non-engagement was most common among older participants, those with an immigrant background, and smokers reporting low levels of alcohol use at baseline [17].

The studies cited above show that socioeconomic and lifestyle factors can influence participation and engagement in digital interventions for addiction prevention and health promotion. However, only a few variables of social inequality were usually collected and examined. The present study aims to extend the existing knowledge by considering a more comprehensive set of socioeconomic and lifestyle factors that influence the accessibility of *SmartCoach*, a digital intervention program for addiction prevention, for young people.

Based on social cognitive theory, the fully automated SMS- and Internet-based intervention program *SmartCoach* targets social skills, substance resistance skills, and self-management skills [18,19]. Program engagement is stimulated by interactive elements such as quiz questions, message and picture contests, and the integration of a friendly competition with rewards in which program users receive credits with each interaction. In a cluster randomized controlled trial, 1759 students from 89 Swiss secondary and upper secondary school classes were used to evaluate the program’s effectiveness [18,20]. Of these, 1473 (83.7%) students with a mean age of 15.4 years participated in the program and the respective study. Adolescents who did not drink in a problematic way and had better educational levels were the ones who used the program the most; on average, program participants reacted to half of the prompted activities [21].

Longer-term results based on an 18-month follow-up showed that, compared to controls, those in the intervention group experienced reduced tobacco-smoking prevalence, however, no effect was observed on at-risk drinking. No significant moderators of the primary outcomes were observed, i.e., effectiveness of the program was independent of age, sex, migration background, and school level.

While the access of young people in these previous efficacy studies [18,20] may have been influenced by the control condition and non-participants were not systematically assessed, the present study was conducted under more everyday conditions and a comprehensive set of socioeconomic variables was collected from participants and non-participants. Thus, for the first time, the study provides well-founded insights into which factors at the level of the school class and which socioeconomic variables at the individual level predict (1) program participation and (2) program use in a digital addiction prevention program.

## 2. Methods

### 2.1. Participants, Setting, and Procedure

Prevention specialists from cooperating regional centers for addiction prevention asked secondary and upper secondary schools in the Swiss cantons of Aargau and Zurich to participate in the current study. The objectives of the program, its implementation, and procedure in the school class were explained to interested teachers. Participating teachers reserved a period of 30–90 min during the usual school lessons for the program implementation.

During these 30–90 min, the present students were (1) introduced to the topic through a short workshop, (2) informed about the program and the accompanying study, and (3) invited to participate in the program. The workshop and information session were led by prevention experts or master students of psychology, who had training on the study and the program to be provided, as well as expertise working with young people and providing preventive interventions. The 20–60 min workshop aimed to arouse the young people’s interest in the topic of stress and provide basic information on its origins. First, the young people had to explain to an alien in a group discussion what they understand by stress. Then, the development of stress as an imbalance of demands and resources was explained. By means of interactive exercises, such as subtracting 7 from 996 several times in a group, stress was deliberately induced and the young people were asked to observe how stress affects their bodies and thoughts. Finally, the young people were asked to describe their stress level in the areas of family, friends, school, and future by means of emojis.

Subsequently, the students were informed about the *SmartCoach* program with the help of an introductory video available at www.smartcoach.info (accessed on 12 July 2023). Students were asked to use their smartphones to complete an online baseline assessment and study registration. Students without a smartphone and students not willing to fill in the online baseline assessment were asked to fill in an anonymous paper–pencil questionnaire for non-participants. This questionnaire included assessment of demographic data, sociodemographic status of the family, health literacy, and reasons for not participating in the online assessment or the *SmartCoach* program. Students with a running smartphone were asked to participate in an anonymous online survey that included assessment of similar data. Subsequently, within this online assessment, students received detailed information on the *SmartCoach* program, the related study, and friendly competition and were invited to participate.

After giving their informed consent, participants were asked to choose a username, to provide their mobile phone number, and to complete additional assessments on stress and social skills which were necessary for the tailoring of the intervention content. Subsequently, the program participants received individually tailored web feedback directly on their mobile phone (see also the section on the intervention program). During the subsequent 4 months, program participants received individually tailored mobile phone-based life skills training. Students without consent were thanked for their information and asked in a free text field why they did not participate in the program.

### 2.2. Intervention Program

The *SmartCoach* (www.smartcoach.info, accessed on 12 July 2023) program provided tailored web-based feedback and text messages to promote self-management skills, social skills, and substance use resistance skills over a period of four months. The intervention elements of the program were based on social cognitive theory [22,23] and life skills training [24]. Individually tailored feedback was provided to program participants immediately after they completed the online baseline assessment within their school classroom. It comprised textual and graphic feedback on general stress, degrees of stress in different domains, personally employed and proposed strategies for coping, and levels of social skills tailored to the individual. Screenshots of the baseline assessment and the web-based feedback are shown in Figure 1.

Following baseline assessment and feedback, participants received between two and four individually tailored text messages per week on their mobile phones for a period of 17 weeks. The fully automated system created and transmitted these messages. For the first seven weeks, the messages emphasized self-management techniques, such as stress management, emotional self-regulation, and anger and frustration management. In weeks 8–12, the messages focused on social skills, such as making requests, declining unreasonable demands, and interacting with new people. The messages during weeks 13–17 emphasized the development of substance use resistance skills, such as the ability to recognize and resist media influences, social norms of alcohol and tobacco use, and the relationships between social and self-management skills and substance use. The messages were individually tailored, based upon data from the baseline assessment and upon text messaging assessments that occurred over the course of the program.

During the four-month coaching phase, participants received a total of 37 text message prompts that invited interaction like replying to quizzes, retrieving media objects, and participating in challenges or contests. To stimulate program engagement, a friendly competition amongst participants was integrated into the program. Within the friendly competition, program participants could collect credits for each interaction, such as participating in quizzes, creating messages or pictures within contests, or accessing video links integrated in text messages. The more credits participants accumulated, the better their chances were of winning one of several prizes that were part of a prize draw (10 prizes totaling CHF 500) after the program’s completion. Participants could access their credit total at any time from their own profile page and compare it to the total of other program participants. Sample screenshots of the program from the coaching phase are shown in Figure 2.

### 2.3. Assessments and Outcomes

Adolescents’ access was considered both at the school class level and at the individual level. At the school class level, data from the class recruitment protocols were used. These were completed separately for each school class by the recruiting professionals and included the following information: number of students present, time of recruitment, duration of the introductory workshop, as well as the educational level of the school class: secondary or upper secondary school.

All students, whether participating in the program or not, were invited to fill in an anonymous survey on socially stratifying factors. The assessments were be based on the PROGRESS framework [25] but were culturally adapted for Swiss adolescents and included the birth country of both parents and the student for the assessment of migration background. Since Swiss people are very similar to Germans and Austrians in terms of language and cultural and socioeconomic background, a further variable was created based on students and their parents’ country of birth, which describes their origin from a non-German-speaking country. Furthermore, we assessed sex, familial socioeconomic status, and health literacy. The latter entails people’s knowledge, motivation, and competences to access, understand, appraise, and apply health information in order to make judgments and make decisions in everyday life [26] and was assessed by a brief version of the German-language European Health Literacy Survey Questionnaire Adapted for Children (HLS-Child-Q15-DE) [27] and the HLS-Child-Q7-DE [28]. The internal consistency (Cronbach’s alpha) of this recently developed 7-item version for the adolescent sample of this study was 0.73 (*n* = 470). Adolescents’ subjective perceptions of familial socioeconomic status were assessed by a brief instrument [29,30] that was recently translated, adapted, and validated for German adolescents [31].

Among students participating in the program, the following health-related variables were additionally assessed within the online baseline survey: perceived stress, social skills, problem drinking, tobacco smoking, and cannabis use. Perceived stress was measured using a single item from the Swiss Juvenir study [32]: “How often have you had the feeling of being overstressed or overwhelmed in the last month?” Participants were asked to indicate their response on a 5-point Likert scale that ranged from “never” to “all the time”. Social skills were assessed by the brief version of the Interpersonal Competence Questionnaire (ICQ-10) [33]. Problem drinking was assessed by the AUDIT-C [34] with a cut-off of ≥5 based on a large German sample [35]. Nicotine or tobacco smoking was assessed using the question “Have you smoked a puff of nicotine-containing cigarettes, e-cigarettes or vapes within the last 30 days?”. Cannabis use was assessed by an item of the HBSC study [36] addressing the number of cannabis consumption days in the last 30 days.

For this study, we defined program participants as students present in the classroom who (1) completed the survey on their smartphone, (2) gave informed consent to participate in the program, and (3) provided the necessary baseline data on stress, social skills, and their mobile phone number to receive the coaching. Non-participants were students who (1) did not participate in either the smartphone survey or the paper–pencil survey, (2) participated in the non-participant paper–pencil survey, (3) participated in the smartphone survey but did not give informed consent to participate in the program, or (4) participated in the smartphone survey and initially provided informed consent to participate but then did not provide the baseline data necessary for program participation.

Program use was operationalized in terms of the total number of interactions with the program that were logged by the system. During the four-month coaching phase, participants received a total of 37 text message prompts that invited interaction like replying to quizzes, retrieving media objects (e.g., videos, web links), and participating in self-challenges or contests within which they could post short messages or photos and vote on other posts. This information was available for each program participant through the log files of the *SmartCoach* system.

### 2.4. Statistical Analysis

We used generalized linear mixed models (GLMMs) to examine predictors of program participation and program use at the individual and school class levels. Within each GLMM, a random intercept for class was modeled. The general procedure for calculating the multilevel models at the class and individual levels was to first test all individual characteristics univariately and to include in the multivariate model only those predictors that were significant (*p* < 0.05, two-tailed) at the univariate level. In a further step, the predictors with the highest *p*-values were removed one by one from the multivariate model until the final solution contained only predictors that significantly contributed to the prediction of program participation and program use, respectively. Regarding the prediction models for program use, we observed signs of overdispersion in the model residuals. Overdispersion occurs when the observed variance of the count outcome is greater than that expected from the Poisson distribution, which assumes that the mean and variance are equal. To account for overdispersion, we chose a negative binomial distribution, which allows greater flexibility in modeling count data with variance greater than the mean. Model diagnostics, including residual plots and goodness-of-fit statistics, were examined to ensure that the assumptions of the negative binomial GLMM were met. All analyses were performed using R version 4.1.2 and GLMMs using the lme4 [37] package. A type I error rate of *p* < 0.05 on two-sided tests was considered statistically significant.

## 3. Results

### 3.1. Program Participants

Recruitment for the *SmartCoach* program was possible in 28 school classes from 11 schools from September 2022 to December 2022. Figure 3 depicts participants’ progression through the study. A total of 476 students were present within 28 school classes. Of these, 440 completed the online survey on their smartphone, 31 completed the non-participant paper–pencil survey, and 6 refused to participate in the survey. A total of 355/476 (74.6%) students consented to participate in the program and the related study. However, among these, another 40 students did not complete baseline assessment or provide their mobile phone number which was necessary for program participation. These students could be described as passive non-consenters. In total, 315 (66.2%) of the present 476 adolescents gave their active consent and provided the necessary information to be included in the program. The comparative analyses regarding program participation in this study are based on 155 non-participants and 315 program participants for whom corresponding socioeconomic information from the baseline survey was available. The analyses regarding program use were based on all 315 program participants, including those 3 participants who actively requested halting program participation.

### 3.2. Predictors of Program Participation

Table 1 displays the percentage of program participants by baseline characteristics on the level of the individual and the school class. None of the individual baseline characteristics significantly predicted program participation. On the level of the school class, the univariate models identified the variables “educational level” and “duration of workshop” as significant predictors of program participation. While 83.8% of the students in upper secondary school classes participated in the program, only 58.9% of the students in secondary school classes did (odds ratio (OR) 4.2, 95% confidence interval (95%-CI) 1.7–10.4, *p* < 0.01). A duration of the introductory workshop on stress of more than 50 min resulted in a participation rate of 79.1%, while only 54.3% participated in school classes with a workshop duration of up to 20 min (OR 3.5, 95%-CI 1.3–9.7, *p* = 0.02).

The variable “workshop duration” was excluded in the selection process for the final multivariate model and only “educational level” was retained for the final prediction model for program participation (OR 4.2, 95%-CI 1.7–10.4, *p* < 0.01, intraclass correlation coefficient (ICC) 0.19, conditional R^2^ 0.28).

### 3.3. Reasons for Not Participating in the Program

Of the 116 students who completed the non-participant survey, 90 provided usable free text information on the reasons for not participating in the *SmartCoach* program. Among the 94 reasons mentioned, the most common was that they had no desire (*n* = 21) or interest (*n* = 4) in participating in the program. Another 22 students said they had no stress at the moment or saw no need to participate. A further five indicated that they had other ways of dealing with stress and four said they were already good at dealing with stress. No time was given as a reason by nine students, while two found the effort to participate in the program too high. Seven people found the questions within the program too personal. Five students each stated that they did not have a smartphone or that its battery was empty. Three persons did not participate for data protection reasons, because they already receive enough messages via smartphone, or are not allowed to register for such programs themselves, respectively. One student found the program too anonymous.

### 3.4. Predictors of Program Use

The mean number of interactions with the SmartCoach program was 15.9 (SD = 14.7), i.e., participants engaged in almost half of the 37 possible interactions with the program. Table 1 displays the mean number of interactions with the *SmartCoach* program by baseline characteristics on the level of the individual and the school class. None of the baseline characteristics on the level of the school class significantly predicted program use. On the level of the individual, the univariate models identified the variables “age”, “migration background”, “origin from a non-German-speaking country”, and “perceived stress” as significant predictors of program use. Compared to the reference category of 14-year-old students, program use was significantly lower for students who were 16 or older (incidence rate ratio (IRR) 0.71, 95%-CI 0.63–0.81, *p* < 0.01). Furthermore, participants with a migration background (IRR 0.72, 95%-CI 0.53–0.98, *p* = 0.04) or an origin from a non-German-speaking country (IRR 0.70, 95%-CI 0.52–0.94, *p* = 0.02) showed significantly lower odds of program use. Finally, students with a medium level of perceived stress showed higher program use, compared to those with a low level of stress (IRR 1.6, 95%-CI 1.0–2.4, *p* = 0.04).

The variables “age”, “migration background”, and “perceived stress” were excluded in the selection process for the final multivariate model and only the variable “origin from a non-German-speaking country” was retained in the final prediction model for program use (IRR 0.70, 95%-CI 0.52–0.94, *p* = 0.02, ICC 0.01, conditional R^2^ 0.04).

## 4. Discussion

### 4.1. Principal Results

Using a proactively recruited sample of adolescents in secondary and upper secondary schools, this study examined a comprehensive set of socioeconomic and other predictors of program participation and program use with a digital life skills intervention program. The main findings are as follows: two out of three students (66%) participated in the program, with upper secondary school students showing significantly higher participation rates (84%) than secondary school students (59%). On average, program participants took part in almost half of the interactions prompted by the SMS text messaging program, with significantly higher program use by students from German-speaking countries of origin.

Through proactive recruitment of students within the school curriculum, the majority of those present were recruited to participate in SmartCoach. The participation rate is thus similar to other interventions where a personal invitation for a mobile phone-based prevention program took place in the classroom [7,9,10,38].

In the *SmartCoach* program, young people were invited to participate following a workshop on stress in the context of their school class. Participants then received individually tailored life skills messages on their personal smartphones for a period of four months. Interestingly, but not unexpectedly, predictors on the level of the school class (duration of introductory workshop or educational level) were associated with program participation whereas predictors on the individual level (age, stress, migration background, or country of origin) were associated with program use. This underlines the importance of class characteristics in recruiting participants and the greater relevance of individual characteristics for program use.

In contrast to several previous studies, we could not identify sex or migration status as significant predictors of program participation [7,9,10,11]. Other socioeconomic factors like health literacy or the familial socioeconomic status did not affect the participation rate either. This finding is in line with the results of a recent Australian study [11] and suggests that those students potentially most in need of health behavior change counseling are just as likely to use the *SmartCoach* program.

On the other hand, the type of school played an important role, and the participation rate in classes from upper secondary schools was significantly higher than in secondary schools. Based on the free text answers, which attribute non-participation mainly to a lack of interest in the topic of stress, a lower level of stress among young people in secondary than upper secondary school classes might be a reason for the lower participation rate in this subgroup. An exploratory comparison of the perceived stress scores between secondary and upper secondary school students of this study revealed that the stress level is significantly higher among the upper secondary school students, which supports this assumption. An increased consideration of stressors relevant for students with lower educational level in the workshops, beyond pressure to perform, might be appropriate to increase interest and participation in the program.

With regard to program use, the results of this study are in line with previous studies that found higher use among younger youth and those without a migration background [16,17,21]. Within this study, which was conducted exclusively in the German-speaking part of Switzerland, it was also shown, however, that the origin of the young person or a parent from a non-German-speaking country of origin was the best predictor of low program use. The reasons for this are probably complex and will be identified in more detail in a forthcoming qualitative study. In addition to language difficulties, cultural differences in particular could lead to individual program contents being less interesting or relevant for young people with a non-German-speaking background.

As the reach across the target population determines the public health impact of an intervention [5] and poor program engagement typically jeopardizes program efficacy [39,40], the main purpose of this study was to identify starting points for program improvement. Population-based mHealth interventions with proactive outreach to a defined target group and automatically generated but individually tailored intervention content, like SmartCoach, provide an excellent opportunity to balance intervention-generated inequalities and increase program participation and use by advanced tailoring of intervention contents. In this sense, the next step will be to particularly involve adolescents from secondary schools and participants with a non-German-speaking background in a qualitative study to identify in more detail factors influencing program participation and use and, based on this, optimize the program by advanced tailoring of intervention contents for these subgroups.

### 4.2. Limitations

The results of this study must be interpreted in light of its limitations. First, they rely on a convenience sample, and the findings might not be generalizable to the adolescent population. Second, this study used self-report data, which carries the potential that social desirability may have affected the findings. Third, the interactions with the *SmartCoach* program were rewarded with credits, and credits were linked to a prize draw. Fourth, the statistical power was limited, particularly for the examined predictor variables on the level of the school class.

## 5. Conclusions

*SmartCoach* is an attractive offer for young people, in which two out of three young people who are invited in the classroom to participate do so. Among the program participants, the use of the program is acceptable, with an average of almost half of the content being worked on. Regarding socioeconomic inequalities, the results do not indicate a negative equity impact of the intervention, as socioeconomic status was neither associated with program participation nor program use. There is potential for improvement in terms of recruitment, especially in school classes with a lower level of education. For these, a less strong focus on performance pressure and a stronger consideration of other stressors could increase the willingness to participate in the program. The most important starting point for improving program use lies in taking greater account of needs and wishes of students with non-German-speaking countries of origin.

## Figures and Tables

**Figure 1 ijerph-20-06379-f001:**
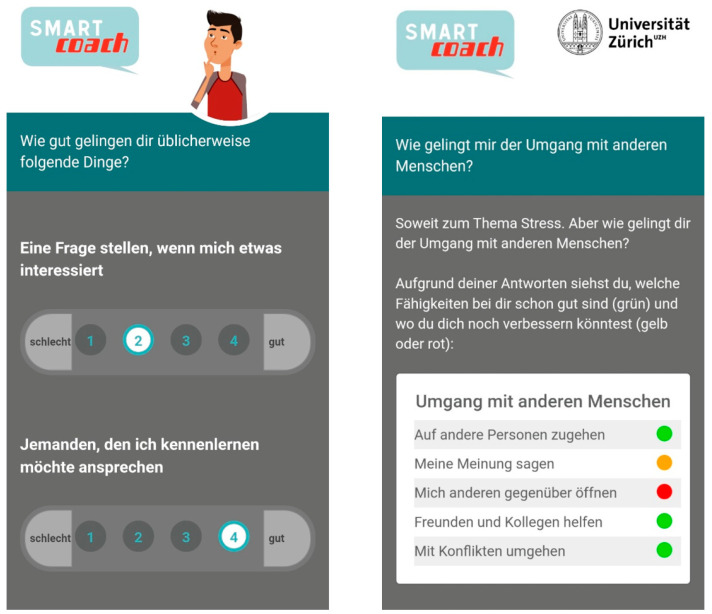
Screenshots from the *SmartCoach* program: (**left**) assessment of social skills; (**right**) feedback on individual social skills.

**Figure 2 ijerph-20-06379-f002:**
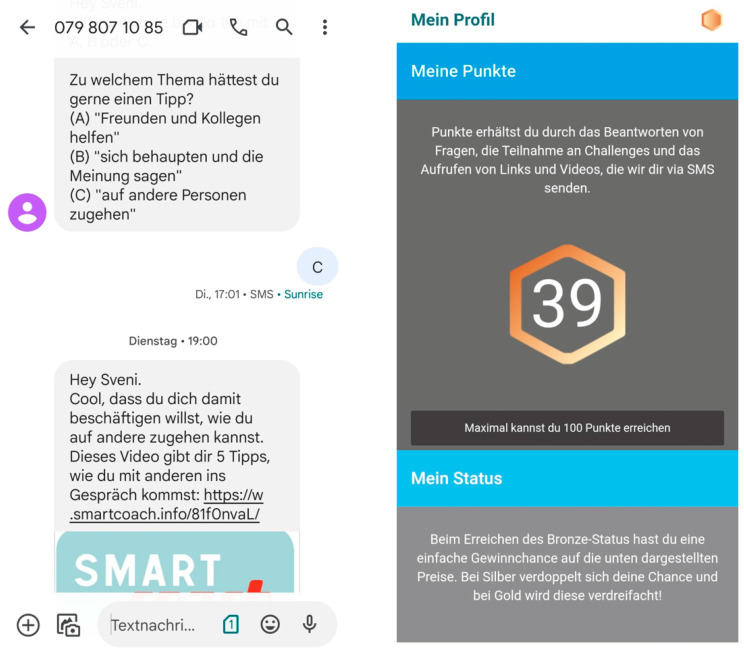
Screenshots from the *SmartCoach* program: (**left**) text messaging-based coaching on social skills; (**right**) individual profile showing the number of credits collected within the friendly competition.

**Figure 3 ijerph-20-06379-f003:**
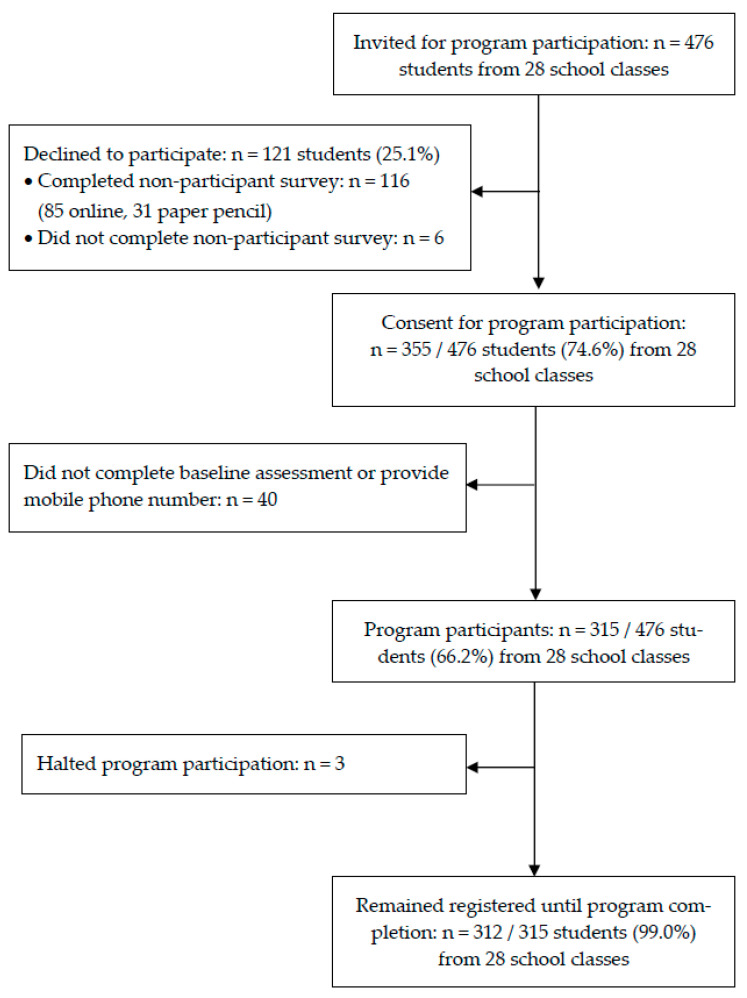
Flow of participants through the study.

**Table 1 ijerph-20-06379-t001:** Program participation and program use (interactions with program) by baseline characteristics on the level of the individual and the school class.

Level/Characteristic	Variable Category	Percentage of Program Participants (*n* = 470)	Mean Number of Interactions with Program (*n* = 315)
**Individual**			
**Sex**	Male	61.4 (223)	13.6 (137)
	Female	72.1 (247)	17.7 (178)
**Age in years**	14	60.8 (125)	**14.9 (76)**
	15	61.3 (181)	17.2 (111)
	16 and older	78.0 (164)	**15.4 (128)**
**Socioeconomic status**	Low	68.9 (132 ^a^)	15.7 (91)
	Medium	68.6 (226)	16.0 (155)
	High	62.7 (110)	15.8 (69)
**Health literacy**	Low	66.7 (147)	16.7 (98)
	Medium	69.4 (144)	15.5 (100)
	High	65.4 179)	15.6 (117)
**Migration background**	No	72.2 (176)	**19.1 (127)**
	Yes	63.9 (294)	**13.8 (188)**
**Origin from a non-German-speaking country**	No Yes	69.4 (209 ^b^) 65.2 (256)	**19.0 (145 ^c^)** **13.2 (167)**
**At-risk alcohol use**	No		15.9 (254)
	Yes		15.7 (61)
**Nicotine/Tobacco smoking**	No		16.1 (232)
	Yes		15.2 (83)
**Cannabis** **use**	No		16.5 (277)
	Yes		11.3 (38)
**Perceived stress**	Low		**12.0 (56)**
	Medium		**18.6 (106)**
	High		15.4 (153)
**Social skills**	Low		17.6 (88)
	Medium		14.1 (97)
	High		16.1 (130)
**School class**			
**Educational level**	Secondary	**58.9 (316)**	14.8 (186)
	Upper secondary	**83.8 (154)**	17.5 (129)
**Time of recruitment**	8 to 9 a.m.	63.0 (165)	15.6 (104)
	10 to 12 a.m.	70.6 (228)	15.4 (161)
	1 to 3 p.m.	64.9 (77)	18.3 (50)
**Duration of workshop**	Up to 20 min	**54.3 (151)**	15.3 (82)
	21–50 min	64.4 (132)	14.3 (85)
	Over 50 min	**79.1 (187)**	17.1 (148)
**Number of students present**	10 to 15	60.3 (136)	13.0 (82)
	16 to 19	72.4 (181)	17.9 (131)
	20 and more	66.7 (153)	15.7 (102)

Notes: Missing values: ^a^ n = 2, ^b^ n = 5, ^c^ n = 3. Variable categories with statistically significant differences (*p* < 0.05) in the univariate prediction models are in bold type. The first category of each variable was the reference for the comparison.

## Data Availability

Data are available on request due to restrictions. The datasets generated and analyzed during the current study are not publicly available due to the Swiss data protection law but are available from the corresponding author on reasonable request. Requests will be reviewed for reasonability and compliance with the study purpose and the participants’ informed consent.

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
