# Peer review of "Predictors of Youth Accessibility for a Mobile Phone-Based Life Skills Training Program for Addiction Prevention"

_ijerph, 2023, doi:10.3390/ijerph20146379_

Round 1

Reviewer 1 Report

This manuscript reports on a programme evaluation that might be interesting for readers depending on their national context. However, the argument is quite consistent and the evidence basis is sound.

In my view, it is almost ready for publication. I suggest the authors to add, if relevant, a few details that might help readers understand their conclusions. 

First, the introduction discusses the criteria to assess the equity of the programme. A brief comment on this issue would greatly enrich the conclusions. 

Second, if I understood correctly, upper secondary education students took more advantage of the programme than lower secondary education students. I wonder if the authors can add minor items of complementary education to clarify whether these age groups are more familiar with different digital tools. Particularly, if any piece of evidence indicates that upper secondary education students use certain tools more often (e.g. SMS, virtual learning environments), it would contribute to contextualise the findings. 

Author Response

This manuscript reports on a programme evaluation that might be interesting for readers depending on their national context. However, the argument is quite consistent and the evidence basis is sound.

In my view, it is almost ready for publication. I suggest the authors to add, if relevant, a few details that might help readers understand their conclusions. 

First, the introduction discusses the criteria to assess the equity of the programme. A brief comment on this issue would greatly enrich the conclusions. 

Reply:

Many thanks for this suggestion. We added a concluding sentence on the equity impact of the program in the Conclusions section: "Regarding socioeconomic inequalities, the results do not indicate a negative equity impact of the intervention, as socio-economic status was neither associated with program participation nor program use."

Second, if I understood correctly, upper secondary education students took more advantage of the programme than lower secondary education students. I wonder if the authors can add minor items of complementary education to clarify whether these age groups are more familiar with different digital tools. Particularly, if any piece of evidence indicates that upper secondary education students use certain tools more often (e.g. SMS, virtual learning environments), it would contribute to contextualise the findings. 

Reply:

We did not find a moderating effect of the school type on the efficacy of the program in our SmartCoach efficacy study

Paz Castro, R., Haug, S., Wenger, A., Schaub, M.P. (2022). Longer-Term Efficacy of a Digital Life-Skills Training for Substance Use Prevention. American Journal of Preventive Medicine, 63(6), 944-953.

However, in the present study, the type of school played an important role concerning program participation.
This issue was addressed in the 5th paragraph of the discussion section: "The participation rate in classes from upper secondary schools was significantly higher than in secondary schools. Based on the free text answers, which attribute non-participation mainly to a lack of interest in the topic of stress, a lower level of stress among young people in secondary than upper secondary school classes might be a reason for the lower participation rate in this subgroup. An exploratory comparison of the perceived stress scores between secondary and upper secondary school students of this study revealed that the stress level is significantly higher among the upper secondary school students, which supports this assumption. An increased consideration of stressors relevant for students with lower educational level in the workshops, beyond pressure to perform might be appropriate to increase interest and participation in the program."

Reviewer 2 Report

The study concerns some aspects of a mobile phone-based training program for addiction prevention. The authors stated in the last introduction paragraph that their study provides “insights into which factors (…) predict (a) program participation and (b) program use in a digital addition prevention program”, and it seems that they are interested in the problem of social inequalities in accessibility for such programs.

I believe that this problem is worthy of exploration, so the results of the study can be appreciated highly.  I read the paper with high interest, nevertheless with mixed feelings. On one hand, a reader got a very detailed description of the program, procedures, materials, variables assessments, etc., so she had every right to expect how successful this program was in the prevention of addiction. But regrettably, there was nothing about such results. But on the other hand, we know that the aim of the paper was to find out what factors were responsible for participation in the program, so omitting such information can be justified. In this case, going into so many details in the program description is pointless, because it generates that expectation as I mentioned above. I suggest that this part of the paper should be made shorter, and limited to necessary information; also, the abstract is too long. In this way, the paper will be more consistent regarding relationships between the purpose of the study and the presentation of the method.

The other way of improving the paper will be placing the detailed method description in a separate file as supplementary materials. Free space, obtained in such a way, could be used to present an analysis of program results.

Minor problems:

--- Please explain what “the RE-AIM framework means” (the second paragraph of the Introduction);

--- Please explain what is the difference between (a) program participation and (b) program use (the last paragraph of the Introduction); since there is no reference to (b) in the Results section.

Author Response

The study concerns some aspects of a mobile phone-based training program for addiction prevention. The authors stated in the last introduction paragraph that their study provides “insights into which factors (…) predict (a) program participation and (b) program use in a digital addition prevention program”, and it seems that they are interested in the problem of social inequalities in accessibility for such programs.

I believe that this problem is worthy of exploration, so the results of the study can be appreciated highly.  I read the paper with high interest, nevertheless with mixed feelings. On one hand, a reader got a very detailed description of the program, procedures, materials, variables assessments, etc., so she had every right to expect how successful this program was in the prevention of addiction. But regrettably, there was nothing about such results. But on the other hand, we know that the aim of the paper was to find out what factors were responsible for participation in the program, so omitting such information can be justified. In this case, going into so many details in the program description is pointless, because it generates that expectation as I mentioned above. I suggest that this part of the paper should be made shorter, and limited to necessary information; also, the abstract is too long. In this way, the paper will be more consistent regarding relationships between the purpose of the study and the presentation of the method.

Reply:

Thank you for this critical comment on the comprehensiveness of the description of the program. As the reviewer correctly describes, the article is not about the effectiveness of the programme but about factors that influence programme participation and use. But particularly against this background, the presentation of the recruitment methodology in the school class via workshop and a precise programme description are important from our point of view, since the programme content and features are decisive for participation in the program and its use.

The other way of improving the paper will be placing the detailed method description in a separate file as supplementary materials. Free space, obtained in such a way, could be used to present an analysis of program results.

Reply:

Thanks for this suggestion. The detailed results concerning program efficacy have been published in separate previous studies.  

Paz Castro, R., Haug, S., Wenger, A., Schaub, M.P. (2022). Longer-Term Efficacy of a Digital Life-Skills Training for Substance Use Prevention. American Journal of Preventive Medicine, 63(6), 944-953.

Haug, S., Paz Castro, R., Wenger, A., Schaub, M.P. (2021). A mobile phone–based life-skills training program for substance use prevention among adolescents: cluster-randomized controlled trial. Journal of Medical Internet Research Mhealth & Uhealth, 9(7):e26951.

To reduce duplication, these main results on program efficacy were only summarized as follows in the introduction section:

"Longer-term results based on 18-months follow up showed that compared to controls, those in the intervention group experienced reduced tobacco-smoking prevalence, however no effect was observed on at-risk drinking. No significant moderators of the primary outcomes were observed, i.e., effectiveness of the program was independent of age, sex, migration background and school level."

Minor problems:

--- Please explain what “the RE-AIM framework means” (the second paragraph of the Introduction);

Reply:

Thanks, we now explained this framework in more detail in the second paragraph of the Introduction: "The RE-AIM (Reach Effectiveness Adoption Implementation Maintenance) framework is a planning and evaluation framework that was developed to help make research findings more generalizable by encouraging scientists and evaluators to balance internal and external validity when developing and testing interventions [5,6]. Based on this framework, major dimensions which determine the public health impact of an intervention and could be affected by intervention generated social inequalities include the reach of the target population and its effectiveness."

--- Please explain what is the difference between (a) program participation and (b) program use (the last paragraph of the Introduction); since there is no reference to (b) in the Results section.

Reply:

Thanks, we now described in more detail the difference between program participation and use in the second paragraph of the Introduction:

"Accessibility or reach of the program includes two major dimensions, which were considered in this study: (1) program participation refers to the act of being involved or enrolled in the program and (2) program use refers to the actual utilization or interaction with the program."

Reviewer 3 Report

Comments to the authors:

The reviewer examined the submitted paper titled “Predictors of youth accessibility for a mobile phone-based life-skills training program for addiction prevention” with great interest. In this paper, the authors focused on participation and persistence rates in SmartCoach, which is a mobile phone-based life skills training program for addiction prevention. Consequently, the authors confirmed that SmartCoach had relatively high of participation and persistence rates and had significant effects on young students in Switzerland. On the other hand, the authors also found that migrant students who originated from non-German-speaking countries were less likely to participate in the program.

Because the authors succeeded in confirming the significant effects of the mobile phone-based program on addiction prevention among young students, the reviewer believes that the submitted paper may contribute to the literature on public health studies. Additionally, they also found a social disparity in using SmartCoach for addiction prevention between students from non-German-speaking countries and other students. Even though the authors did not examine the background of the differences in using the program between students in detail, this finding is very interesting for understanding social inequality in mental health between migrants and non-migrants. It can be said that the authors present a new task for social researchers interested in social inequality studies.

However, the reviewer has concerns regarding the submitted manuscript. He believes that the authors should appropriately address these issues for publication in the International Journal of Environment Research and Public Health. If the authors think that they do not need to address them, the reviewer will request that they explain why they think so. After suitably addressing or responding to the reviewer’s concerns, the reviewer could recommend the submitted paper for publication.

First, the authors did not seem to argue for the effects of SmartCoach itself on addiction prevention. The authors examined only the participation and persistence rates in detail but did not examine the extent of its effects on addiction prevention. However, if SmartCoach had only low effects on addiction prevention, its participation and persistence rates would lose significance in public health studies. Therefore, the reviewer believes that the authors should discuss not only the participation and persistence rates, but also the program’s effects on addiction prevention.

Second, the study clarified that migrant students from non-German-speaking countries were less likely to participate in SmartCoach. This finding implies that they suffered from social disparities in life skills due to losing the opportunity to participate in an effective program. The reviewer believes that such widening social disparities among migrant students and non-migrant students due to technological innovation may be a serious problem that needs to be addressed by social researchers. Therefore, the reviewer expects the authors to examine the problem more clearly as a dark side of technological innovation in the field of education.

Finally, although the authors referred to the analytical results of the multivariate models, they did not report them correctly in the submitted paper. If they refer to the analytical results of the multivariate models, they should present a regression table. However, the authors did not present one and avoided explaining the results in detail. Therefore, the reviewer expects that the authors create and present a regression table in the revised paper.

The reviewer appreciates the opportunity to review this paper. He sincerely hopes that his comments will contribute to the improvement of this paper.

Author Response

The reviewer examined the submitted paper titled “Predictors of youth accessibility for a mobile phone-based life-skills training program for addiction prevention” with great interest. In this paper, the authors focused on participation and persistence rates in SmartCoach, which is a mobile phone-based life skills training program for addiction prevention. Consequently, the authors confirmed that SmartCoach had relatively high of participation and persistence rates and had significant effects on young students in Switzerland. On the other hand, the authors also found that migrant students who originated from non-German-speaking countries were less likely to participate in the program.

 Because the authors succeeded in confirming the significant effects of the mobile phone-based program on addiction prevention among young students, the reviewer believes that the submitted paper may contribute to the literature on public health studies. Additionally, they also found a social disparity in using SmartCoach for addiction prevention between students from non-German-speaking countries and other students. Even though the authors did not examine the background of the differences in using the program between students in detail, this finding is very interesting for understanding social inequality in mental health between migrants and non-migrants. It can be said that the authors present a new task for social researchers interested in social inequality studies.

However, the reviewer has concerns regarding the submitted manuscript. He believes that the authors should appropriately address these issues for publication in the International Journal of Environment Research and Public Health. If the authors think that they do not need to address them, the reviewer will request that they explain why they think so. After suitably addressing or responding to the reviewer’s concerns, the reviewer could recommend the submitted paper for publication.

First, the authors did not seem to argue for the effects of SmartCoach itself on addiction prevention. The authors examined only the participation and persistence rates in detail but did not examine the extent of its effects on addiction prevention. However, if SmartCoach had only low effects on addiction prevention, its participation and persistence rates would lose significance in public health studies. Therefore, the reviewer believes that the authors should discuss not only the participation and persistence rates, but also the program’s effects on addiction prevention.

 Reply:

The detailed results concerning program efficacy have been published in separate previous studies.  

Paz Castro, R., Haug, S., Wenger, A., Schaub, M.P. (2022). Longer-Term Efficacy of a Digital Life-Skills Training for Substance Use Prevention. American Journal of Preventive Medicine, 63(6), 944-953.

Haug, S., Paz Castro, R., Wenger, A., Schaub, M.P. (2021). A mobile phone–based life-skills training program for substance use prevention among adolescents: cluster-randomized controlled trial. Journal of Medical Internet Research Mhealth & Uhealth, 9(7):e26951.

To reduce duplication, these main results on program efficacy were only summarized as follows in the introduction section:

"Longer-term results based on 18-months follow up showed that compared to controls, those in the intervention group experienced reduced tobacco-smoking prevalence, however no effect was observed on at-risk drinking. No significant moderators of the primary outcomes were observed, i.e., effectiveness of the program was independent of age, sex, migration background and school level."

Second, the study clarified that migrant students from non-German-speaking countries were less likely to participate in SmartCoach. This finding implies that they suffered from social disparities in life skills due to losing the opportunity to participate in an effective program. The reviewer believes that such widening social disparities among migrant students and non-migrant students due to technological innovation may be a serious problem that needs to be addressed by social researchers. Therefore, the reviewer expects the authors to examine the problem more clearly as a dark side of technological innovation in the field of education.

 Reply:

Thanks for this interpretation. As the variables "socioeconomic status", "educational level" and "social skills" did not predict program use, we do not believe that social disparities in life skills are responsible for the differences in program use. As described in the Discussion, the reasons for this finding are probably complex and require further investigation using qualitative methods:

"With regard to program use, the results of this study are in line with previous studies that found higher use among younger youth and those without a migration background [16,17,21]. Within this study, which was conducted exclusively in the German-speaking part of Switzerland, it was also shown, however, that the origin of the young person or a parent from a non-German-speaking country of origin was the best predictor of low program use. The reasons for this are probably complex and will be identified in more detail in a forthcoming qualitative study. In addition to language difficulties, cultural differences in particular could lead to individual program contents being less interesting or relevant for young people with a non-German-speaking background."

Finally, although the authors referred to the analytical results of the multivariate models, they did not report them correctly in the submitted paper. If they refer to the analytical results of the multivariate models, they should present a regression table. However, the authors did not present one and avoided explaining the results in detail. Therefore, the reviewer expects that the authors create and present a regression table in the revised paper.

Reply:

The final prediction model for program participation solely included "educational level" and the final model for program use solely included "origin from a non-German speaking country". Therefore, the final regression models were easier and clearer to present in the text. We added the detailed results of these final prediction models in the respective paragraphs of the Results:

"The variable "workshop duration" was excluded in the selection process for the final multivariate model and only "educational level" was retained for the final prediction model for program participation (OR 4.2, 95%-CI 1.7-10.4, p <.01, Intra-class Correlation Coefficient (ICC) 0.19, Conditional R2 0.28)."

The variables "age", "migration background" and "pereceived stress" were ex-cluded in the selection process for the final multivariate model and only the varia-ble "origin from a non-German speaking country" was retained in the final predic-tion model for program use (IRR 0.70, 95%-CI 0.52-0.94, p=.02 , ICC 0.01, Conditional R2 0.04). 

The reviewer appreciates the opportunity to review this paper. He sincerely hopes that his comments will contribute to the improvement of this paper.